# Increased Bone Marrow Uptake and Accumulation of Very-Late Antigen-4 Targeted Lipid Nanoparticles

**DOI:** 10.3390/pharmaceutics15061603

**Published:** 2023-05-27

**Authors:** Laura E. Swart, Marcel H. A. M. Fens, Anita van Oort, Piotr Waranecki, L. Daniel Mata Casimiro, David Tuk, Martijn Hendriksen, Luca van den Brink, Elizabeth Schweighart, Cor Seinen, Ryan Nelson, Anja Krippner-Heidenreich, Tom O’Toole, Raymond M. Schiffelers, Sander Kooijmans, Olaf Heidenreich

**Affiliations:** 1Princess Máxima Center for Pediatric Oncology, Heidelberglaan 25, 3584 CS Utrecht, The Netherlands; 2Department of Pharmaceutics, Utrecht Institute for Pharmaceutical Sciences, Utrecht University, 3584 CG Utrecht, The Netherlands; 3CDL Research, University Medical Center Utrecht, Heidelberglaan 100, 3584 CX Utrecht, The Netherlands; 4Wolfson Childhood Cancer Research Centre, Newcastle University, Newcastle upon Tyne NE1 7RY, UK

**Keywords:** acute myeloid leukemia, targeted delivery, siRNA lipid nanoparticles (LNPs), very-late antigen-4 (VLA-4), bone marrow targeting

## Abstract

Lipid nanoparticles (LNPs) have evolved rapidly as promising delivery systems for oligonucleotides, including siRNAs. However, current clinical LNP formulations show high liver accumulation after systemic administration, which is unfavorable for the treatment of extrahepatic diseases, such as hematological disorders. Here we describe the specific targeting of LNPs to hematopoietic progenitor cells in the bone marrow. Functionalization of the LNPs with a modified Leu-Asp-Val tripeptide, a specific ligand for the very-late antigen 4 resulted in an improved uptake and functional siRNA delivery in patient-derived leukemia cells when compared to their non-targeted counterparts. Moreover, surface-modified LNPs displayed significantly improved bone-marrow accumulation and retention. These were associated with increased LNP uptake by immature hematopoietic progenitor cells, also suggesting similarly improved uptake by leukemic stem cells. In summary, we describe an LNP formulation that successfully targets the bone marrow including leukemic stem cells. Our results thereby support the further development of LNPs for targeted therapeutic interventions for leukemia and other hematological disorders.

## 1. Introduction

Chromosomal rearrangements are a hallmark of pediatric acute myeloid leukemias (AMLs). These rearrangements give rise to leukemic fusion genes that initiate and drive leukemia. Therefore, they present ideal therapeutic targets; however, targeting fusion genes using conventional drug molecules has proven challenging. A promising alternative is to target fusion transcripts by RNA interference [1,2,3,4,5]. Because of the poor pharmacokinetic properties of siRNAs, caused by low stability, poor uptake by cells, fast clearance and induction of immunogenic responses, delivery vehicles such as micelles, liposomes, nanoplexes, or lipid nanoparticles (LNPs) are required [6,7,8,9,10,11,12,13,14,15]. LNPs are currently amongst the most promising non-viral delivery systems used for the delivery of siRNAs because of their high encapsulation efficiency combined with reduced immunogenicity and improved circulation times [14]. Commonly, LNPs consist of a cationic ionizable lipid such as D_lin_-MC_3_-DMA or C12-200, together with helper lipids distearoylphosphatidylcholine (DSPC) and cholesterol. To provide stealth properties, lipids conjugated to polyethylene glycol (PEG) are added to the formulation, such as 1,2-dimyristoyl-rac-glycero-3-methoxyPEG-2000 (DMG-PEG_2000_) or 1,2-Distearoyl-sn-glycero-3-phosphoethanolamine-PEG-2000 (DSPE-PEG_2000_).

The majority of systemically administered LNPs accumulate in the liver due to the liver’s large size, functionalized vascular structure and, for some LNP types, surface-absorbed apolipoprotein-E-mediated uptake by the hepatocytes [16,17,18]. The resultant liver accumulation is widely exploited to effectively modulate therapeutic targets in this organ. Targeting other organs and tissues has proven to be more challenging [17,19,20]. The bone marrow, for instance, is the site where most hematological diseases, including leukemia, originate. In principle, the bone marrow is accessible for LNPs as it is well vascularized with sinusoids that have highly fenestrated endothelial layers required for the facile migration of mature cells into the bloodstream [21]. Consequently, these sinusoids might also allow access by lipid nanocarriers. However, current retention times of LNPs in the bone marrow are too low to reach therapeutically effective doses for therapeutic oligonucleotides. To increase tissue retention, the physicochemical properties of the LNPs can be modified, which is also known as passive targeting. For drug delivery, functionalizing LNPs with chemical or biological moieties can significantly increase LNP cellular specificity [22,23,24,25,26]. The targeting moiety should ideally bind to a receptor that is disease-specific, highly expressed on target cells and induce receptor-mediated endocytosis of the LNPs after binding.

Integrin receptors display all these qualities. These heterodimeric cell adhesion receptors are involved in cell–cell and cell–extracellular matrix interactions. Several integrin dimers such as the very-late antigen-4 (VLA-4) receptor, the lymphocyte function-associated antigen-1 and the Lymphocyte Peyer patch adhesion molecule have been explored for targeting hematological malignancies [27,28,29,30,31,32,33,34]. VLA-4 is expressed on all leukocytes and plays a key role in mediating homing to and retention of hematopoietic stem and progenitor cells (HSPC) in the bone marrow [32,35]. It is a heterodimer of integrin α4 (CD49d) and β1 (CD29b) and binds to vascular cell adhesion molecule-1 (VCAM-1) present on endothelial cells and fibronectin. Because of its crucial role in the pathophysiology of many diseases, this receptor is a strong candidate as a therapeutic target. The tripeptide Leu-Asp-Val (LDV) was identified as the smallest sequence of fibronectin necessary to bind the VLA-4 receptor [36]. The terminal amino acid of this tripeptide is chemically modified with a benzyloxycarbamido phenylurea group to increase the binding potency, offer protection for enzymatic hydrolysis and aid in T cell adhesion inhibition [37,38,39].

We hypothesized that targeting the VLA-4 receptor using LDV-functionalized LNPs potentially leads to increased uptake and retention of LNPs in the bone marrow by interacting with resident leukocytes. We, therefore, investigated the uptake and efficacy of LDV-LNPs in patient-derived xenotransplants (PDX) and their biodistribution in mice. LDV-LNPs showed improved cellular uptake and efficacious siRNA delivery in PDX, in contrast to their non-targeted counterparts. In vivo, LDV-LNPs displayed a significantly higher accumulation in the bone marrow compared to non-targeted LNPs and an increased association of LDV-LNPs with both immature and mature hematopoietic bone marrow cells. These data provide proof of concept for the delivery of siRNA to the bone marrow through VLA-4 targeting, which could be used to treat a broad range of hematological disorders, including leukemia.

## 2. Material and Methods

### 2.1. siRNA LNPs Preparation

LNPs were prepared as described previously [40]. Briefly, a 25 mM lipid mix was prepared by dissolving D_Lin_-MC3-DMA:DSPC:Cholesterol:DMG-PEG_2000_ in absolute ethanol at molar ratios of 50:10:38.5:1.5. To specifically target the leukemic fusion gene *RUNX1/ETO*, we designed an siRNA homologous to the fusion site of the *RUNX1/ETO* transcript: siRE-mod. As a mismatch control, we used siMM-mod, where two nucleotides in the siRE sequence have been swapped [5,41]. siRNAs were dissolved in a hybridization buffer (25 mM HEPES, 100 mM NaCl, pH 7.5) to a final concentration of 100 µM. To generate LNPs with an optimal nitrogen/phosphate ratio of 4 [42], siRNAs were diluted to a concentration of 26 µM in 25 mM acetic acid pH 4. The siRNA sequences are included in Appendix A. For ex vivo visualization, 0.1% molar ratio Cy3-DBCO-DSPE-PEG_2000_ was post-inserted into preformed LNPs. For in vivo visualization, 50% or 30% of the siRE-mod was replaced by siRE-mod-Cy7 (Axolabs, Kulmbach, Germany), conjugated to the sense strand and 0.2% molar ratio DSPE-Cy5.5 was added as a lipid label. siRNA LNPs were prepared using microfluidic mixing in a NanoAssemblr Benchtop Instrument (Precision Nanosystems, Vancouver, Canada) with a total flow rate of 4 mL/min and 1:3 lipid:siRNA volume ratio. Ethanol and acetic acid were removed by overnight dialysis against phosphate-buffered saline (PBS) at 4 °C using sterile 10 kDa molecular weight cut-off (MWCO) Slide-A-Lyzers (Thermofisher Scientific^TM^, Waltham, MA, USA) with PBS replacements after one and three hours. After dialysis, the LNPs were concentrated using Amicon^®^Ultra-4 Centrifugal Filter Units with 10 kDa MWCO cellulose membrane (Amicon^®^, Merck, Rahway, NJ, USA) according to the supplier’s protocol.

### 2.2. Particle Size Determination by Dynamic Light Scattering (DLS)

The hydrodynamic diameter and polydispersity index (PDI) of 1:10 diluted samples in Dulbecco’s phosphate-buffered saline (DPBS) were measured before and after post-insertion of ligand-conjugates using the Zetasizer Nano Series ZS (Malvern Instruments, Malvern Panalytical, Malvern, UK). Samples were measured in three runs over 5 min at 25 °C, with backscatter set at 173° and attenuator set at 7. Hydrodynamic diameters and PDI values were averaged from three runs.

### 2.3. Zeta Potential Determination

The zeta potential of the LNPs was determined using the Zetasizer Nano Series ZS. LNPs were diluted 1:100 in 10 mM HEPES (pH = 7.4) prior to analysis. Each sample was measured three times.

### 2.4. Particle Size Determination by Cryo-Electron Microscopy (cryoEM)

The LNPs were characterized by cryoEM as previously described [40].

### 2.5. siRNA Concentration and Encapsulation Efficiency

Total siRNA amount loaded into the LNPs was determined using the Quant-It^TM^ Ribogreen microRNA Assay kit (Thermofisher Scientific^TM^) in the presence of 0.5% Triton X-100, whereas the free/unencapsulated siRNA amount was determined in DPBS. The encapsulation efficiency was calculated by the following formula (([siRNA_triton_] − [siRNA_DBPS_])/[siRNA_triton_]) × 100%.

### 2.6. Click Chemistry Conjugation and Functionalization of LNPs

Functionalized LNPs were prepared following the protocol we previously established [40]. In brief, Cyanine 3-azide (Cy3) or LDV-azide was first conjugated to the ring-constrained alkyne dibenzocyclooctyne (DBCO), which is covalently linked to DSPE-PEG_2000_, overnight at room temperate at a 1:3 molar ratio. The conjugated product was immediately post-inserted at a 0.1% molar ratio into preformed LNPs for 30 min at 45 °C [40]. Functionalized LNPs were kept at 4 °C protected from light for up to 7 days.

### 2.7. Cell Culture

Kasumi-1 (DSMZ no. ACC 220) cell line was obtained from the DSMZ (LGC Standards GmbH, Wesel, Germany) and cultured in Roswell Park Memorial Institute Medium 1640 supplemented with Glutamax (Gibco, Thermofisher Scientific^TM^) 10% fetal bovine serum (FBS, Bodinco, Alkmaar, The Netherlands). Cells were regularly authenticated and tested negative for mycoplasma. Mesenchymal stem cells (MSCs) were obtained from healthy human bone marrow and cultured in Dulbecco’s Modified Eagle Medium low glucose (Gibco) supplemented with 1% L-glutamine, 8 ng/mL recombinant human fibroblast growth factor-basic (Peptrotech, Rocky Hill, CT, USA), 20% FBS and 100 U/mL penicillin–streptomycin (P/S) at a density of 7500 cells/cm^2^ one day prior to PDX seeding [43]. Bone marrow and ascites PDX expressing the *RUNX1/ETO* fusion were co-cultured on MSCs feeders and cultured in serum-free AML expansion medium (Serum-Free Expansion Medium II (Stemcell, Vancouver, Canada) supplemented with 150 ng/mL stem cell factor, 100 ng/mL thrombopoietin, 10 ng/mL Fms-related tyrosine kinase 3 ligand and 1.35 µM UM729, 750 nM StemRegenin 1 (Biogeme, Lausanne, Switzerland), 10 ng/mL interleukin-3, 10 ng/mL granulocyte-macrophage colony-stimulating factor; all cytokines have been purchased from PeptroTech) and 100 u/mL P/S. All patients gave written consent for the use of their material for research purposes. All cells were cultured at 37 °C in a humidified atmosphere containing 5% CO_2_.

### 2.8. LNP Treatment

Leukemia blasts were seeded at a density of 10^6^ cells/mL in AML expansion medium on MSC feeders and treated with 4 ug/mL LNPs or LDV-LNPs for 24 h followed by dilution to a density of 5 × 10^5^ cell/mL allowing expansion. Leukemic cells were collected after 3 days for analysis.

### 2.9. LNP Uptake and Internal ETO Visualization

Cells were seeded at a density of 5 × 10^5^ cells/mL and 100 µL/well in a flat-bottom 96-well plate. Cy3-labeled siRE LNPs with and without LDV-ligand were added to the wells to a final siRNA concentration of 2 (cell lines) or 4 (blasts) µg/mL followed by incubation for up to 24 h at 37 °C and 5% CO_2_. Cells were collected after indicated time points for microscopy and flow cytometry analysis and washed once in sterile FACS buffer (PBS containing 0.025% bovine serum albumin, 0.02% sodium azide and 1% FBS) followed by centrifugation for 4 min at × 350 g (cell lines) or 4 min × 500 g (blasts). Cell pellets were resuspended in 150 µL FACS buffer and washed once with acetic acid buffer (0.5 M NaCl, 0.2 M acetic acid, pH 4) to remove membrane-bound LNPs and sterile PBS. To visualize the uptake via flow cytometry, leukemia blasts were stained with 1:50 huCD90-BV421 (clone 5E10, cat #562556, BD Biosciences, San Jose, CA, USA), 1:50 huCD34-APC (clone 8G12, cat #345804, BD Biosciences, San Jose, CA, USA) and 1:25 hu-IgG in FACS buffer, fixed at least overnight using 2% paraformaldehyde in PBS and fluorescence acquired on a Cytoflex LX. To visualize the uptake and internal ETO expression via microscopy, cells and blasts were spun on coverslips using a Thermofisher Scientific^TM^ Cytospin 4 Centrifuge at program 1 (680 rpm for 7 min) with low acceleration. Subsequently, cells and blasts were permeabilized using 0.1% Triton X-100 for 15 min and counterstained with 4′,6-diamidino-2-phenylindole (DAPI) for 5 min or 1:250 polyclonal anti-RUNX1T1 (cat #PA5-79943, Thermofisher Scientific^TM^) for 1 h followed by 1:2000 gt-anti-rb-IgG-AF555 antibody incubation (cat #405324, Biolegend, San Diego, CA, USA) for 1 h. LNP uptake and internal ETO stain were then determined using a DM6 widefield microscope (Leica Microsystems, Wetzlar, Germany) or an SP8 confocal microscope (Leica Microsystems, Wetzlar, Germany).

### 2.10. RNA Extraction, cDNA Synthesis and qPCR

Total RNA was extracted using the RNeasy Mini Kit (Qiagen, CA, USA) following the supplier’s protocol. A total of 500 ng cDNA was prepared using the RevertAid H Minus First Strand cDNA Synthesis Kit (Thermofisher Scientific^TM^) according to the manufacturer’s protocol. The qPCR primers are provided in Appendix A. cDNA samples were diluted to 15 ng input/well prior to qPCR analysis with RNase-free H_2_O. The SSoAdvanced^TM^ Universal SYBR^®^ Green Supermix (Bio-rad, Berkeley, CA, USA) was used according to the supplier’s protocol.

### 2.11. Protein Extraction and Western Blot for RUNX1/ETO

Proteins were isolated simultaneously with the RNA by precipitating the RNeasy Mini Kit flow-through with 2 volumes of cold acetone and incubation for 2 h at −80 °C. The protein pellets were then dissolved in 9 M Urea buffer (4% CHAPs and 1% dithiothreitol) at a concentration of 5 × 10^4^ cells per µL. Western blotting was carried out as previously described [5]. Polyclonal Rb-anti-RUNX1 (1:250, cat #4334S, Cell Signaling) and 1:10,000 polyclonal m-anti-GAPDH (cat #AM4300, Invitrogen, Carlsbad, CA, USA) were used as primary antibodies. Goat-anti-mouse (1:10,000, cat #P0447, Agilent, Santa Clara, CA, USA) or anti-rb (1:10,000, cat #sc-2004, Santa Cruz Biotechnology, Dallas, TX, USA) polyclonal IgG HRP-conjugates were used as secondary antibodies.

### 2.12. Animal Experiments

All animal experiments were performed with the permission of the Animal Welfare Body Utrecht and complied with the Dutch Experiments on Animals Act (WOD) under license AVD10800202115026. The experiment was carried out in accordance with the Guide for the Care and Use of Laboratory Animals. Animals received ad libitum standard chow and water and were housed under standard conditions with 12 h light/dark cycles until experimental procedures.

### 2.13. In Vivo Biodistribution of LDV-LNPs and LNPs

The biodistribution of LDV-LNPs and LNPs was assessed in female C57/Bl6J (*n* = 8, weight between 18 and 22 g, 11 weeks old, Charles River, Leiden, the Netherlands) upon tail vein injection of LNPs loaded with Cy7-labelled siRNA (50% of total siRNA content). Animals were randomized into groups of three mice receiving either 50 µg LDV-LNPs or undecorated LNPs. Two control mice received PBS. After 2, 4 and 24 h the animals were live-imaged under isoflurane anesthesia.

### 2.14. In Vivo Circulation Time and Uptake of LDV-LNPs and LNPs by Bone Marrow Cell Populations

The biodistribution and uptake of LDV-LNPs and LNPs in bone marrow cell populations were assessed in female BALB/c AnNCrl mice (*n* = 16, weight between 18 and 22 g, 11 weeks old, Charles River, Leiden, the Netherlands) upon intravenous (i.v.) injection of Cy5.5-labelled LNPs (containing a 0.2% molar ratio of DSPE-Cy5.5) loaded with Cy7-labelled siRNA (30% of total siRNA content). Animals were randomized into groups of six mice receiving either LDV-LNPs or non-targeted LNPs. Four control mice received PBS. Either 50 µg of LNPs in 100 µL or an equal volume of vehicle were administered via the tail vein. Blood was collected after 1 min and 1, 2, 4 and 24 h via vena saphena puncture (1 min and 1 and 2 h), submandibular puncture (4 h) and retro-orbital puncture (endpoints) and collected in EDTA anti-coagulated capillaries (1 min and 1 and 2 h) and tubes (endpoints). Blood samples were centrifuged at 2000× *g* for 10 min at 4 °C. Plasma was collected and stored at −80 °C until further analysis.

After 4 (*n* = 4 for LDV-LNPs and LNPs or *n* = 2 for PBS) or 24 h (*n* = 2 for all groups), mice received an intraperitoneal injection of 100 mg/kg of ketamine and 10 mg/kg of xylazine followed by perfusion with PBS via the left ventricular cavity of the heart. Organs (brain, liver, spleen, kidneys, lungs, heart, femora and tibiae) were collected. Next, whole organ tissue distribution of the LNPs was measured using a Pearl Impulse Imager (Li-Cor Biosciences, Lincoln, NE, USA). Fluorescence in plasma samples was measured on a Spectramax ID3 plate reader (Molecular Devices, San Jose, CA, USA) at excitation/emission wavelengths of 680/720 (Cy5.5) and 750/790 (Cy7). Plasma samples were first diluted 1:1 with pooled control mouse plasma (*t* = 1 min and 1 h) and subsequently diluted 1:1 with DPBS. Diluted plasma samples (20 µL) were transferred to a clear-bottom black 384-well plate in duplicates and fluorescence was measured. Data are expressed as a percentage of the fluorescence signal obtained at *t* = 1 min (100% of injected dose).

### 2.15. Flow Cytometry

To study the association of LNPs with femur cells, we isolated single cells from the femur. Briefly, femora were sterilized with 70% ethanol and washed twice with PBS. Both epiphyses were removed and 1 mL of Iscove’s Modified Dulbecco’s Medium (IMDM, Gibco, Carlsbad, CA, USA) was slowly flushed through the length of the bone marrow at each end. Harvested cells were re-suspended and transferred through a 70 µM cell strainer, followed by a wash with 5 mL IMDM.

For staining, single cell suspensions were spun for 5 min at 500× *g* and subsequently re-suspended in 1 mL of Hybri-Max Red Blood Cell (RBC) Lysis Buffer (cat #11814389001, Sigma Aldrich, St. Louis, MO, USA). The suspension was gently mixed for 1 min at room temperature after which 20 mL of PBS was added to deactivate the RBC lysis buffer. Next, cells were pipetted through a 70 µM cell strainer, washed with 10 mL PBS and re-suspended in 5 mL PBS. Cells were counted 1:1 with 0.4% Trypan Blue and seeded at a density of 0.2 × 10^6^–1 × 10^6^ cells/well in a V-bottom 96-well plate, blocked for non-specific binding of immunoglobulin to the Fc receptors on ice for 10 min with Trustain FcX (1:250 in PBS; clone 93, cat# 101319 and spun for 5 min at 500× *g*. The following antibody cocktail was added: 1:200 mCD45-PacificBlue (clone S18009F, cat# 157211), 1:50 mCD34-PE (clone SA376A4, cat# 152203), 1:200 mCD49d-APC (clone R1-2, cat# 103621) and 1:100 Zombie Green^TM^ Fixable Viability Kit (cat# 423111) in 50 µL Trustain FcX 1:250 in PBS; all antibodies were obtained from BioLegend. After incubation for 45 min on ice, cells were washed twice with FACS buffer and then fixed in 100 µL 2% PFA in PBS. Samples were stored in the dark at 4 °C for 1–2 days.

Samples were analyzed with a Cytoflex LX (Beckman Coulter, Brea, CA, USA). Data were analyzed using FlowJo.v10.7.1. Cell populations were identified according to the gating strategies shown in Appendix A.

### 2.16. Statistical Analysis

Data were analyzed with GraphPad Prism 8 (GraphPad Software, Inc., San Diego, CA, USA) using a two-sided unpaired or paired Student’s *t*-test. Differences with *p* values < 0.05 were considered statistically significant.

## 3. Results

### 3.1. Characterization of LNPs and LDV-LNPs

For siRNA delivery to leukemic cells, we packaged modified siRNAs into LNPs containing the cationic ionizable lipid Dlin-MC3-DMA by microfluidic mixing (Figure 1a1). To improve uptake in leukemia cells, we conjugated LDV-azide to DSPE-PEG_2000_ by click-chemistry followed by post-insertion of the LDV-DSPE-PEG_2000_ conjugate into preformed LNPs (Figure 1a2) [40]. The LDV-peptide binds with high affinity to the VLA-4 receptor expressed on the cell surface of hematopoietic and leukemic cells (Figure 1a3), upon uptake of the cargo *RUNX1/ETO* target mRNA in leukemic cells is degraded by RNAi (Figure 1a4). To monitor the biodistribution, we incorporated DSPE-Cy5.5 in LNPs and loaded them with 30% siRNA-Cy7. Both LDV-conjugate post-inserted LNPs or LNPs containing fluorescent-labeled siRNA and DSPE-Cy5.5 displayed similar physicochemical characteristics with a hydrodynamic diameter <100 nm (Table 1), PDI of 0.2 and slightly negative zeta potential (Table 1). CryoEM imaging confirmed that the LNPs and LDV-LNPs are uniformly sized (Figure 2b).

### 3.2. In Vitro Cell Specificity and Delivery Efficacy of LDV-LNPs

To investigate the uptake kinetics of LDV-LNPs in AML cells, we labeled LNPs with Cy3 conjugated to DSPE-PEG_2000_ and monitored the uptake by fluorescence microscopy and flow cytometry. Incorporation of the LDV peptide in LNPs increased their uptake 10-fold in VLA-4-expressing AML cell lines over 24 h (Figure 1c) [40]. Next, we examined the uptake of LDV-LNPs in a more complex cellular environment where *RUNX1/ETO*-expressing PDX are cultivated on MSCs (Figure 1d). The uptake of LDV-LNPs by *RUNX1/ETO*-expressing PDX was increased 10-fold compared to untargeted LNPs within 24 h and led to a twofold reduction of the *RUNX1/ETO* transcript and a complete loss of the fusion protein in different PDX samples after 3 days compared to the controls (Figure 1e–g, Appendix A). The knockdown of RUNX1/ETO on protein level after LDV-LNPs treatment in *RUNX1/ETO*-positive PDX was confirmed by imaging upon intracellular staining of ETO (Figure 1h). An overlay of the DAPI and ETO channels shows nuclear localization of the ETO protein in the control PDX. This fluorescence signal was lost in the treated PDX, indicative of substantial loss of the RUNX1/ETO fusion protein.

### 3.3. Pharmacokinetics and In Vivo Biodistribution of LDV-LNPs and LNPs

Our data demonstrate the benefit of LDV-functionalization for LNP uptake and siRNA delivery efficacy ex vivo. To gain insight into the pharmacokinetics and biodistribution of the LNPs in vivo, we prepared dual fluorescently labeled LNPs containing Cy7-siRNA and DSPE-Cy5.5. LNPs were injected into the tail vein of wild-type C57/Bl6J or BALB/cAnNCrl mice. Next, we performed in vivo imaging and quantified the tissue distribution of both LNP vehicles and siRNA cargo by fluorescent imaging of blood samples and organs (Figure 2a). Treatment did not cause any adverse effect in either mouse strain during the observation times of up to 2 days. In vivo imaging of C57Bl/6J mice showed the highest siRNA Cy7 fluorescent signal in the liver region after 4 h, with a strong reduction in the signal after 24 h (Figure 2b). This was in line with the circulation time of the LNPs, which was analyzed in plasma samples at various time points in BALB/c mice. Both LNPs displayed circulation half-lives of 45 min, with less than 10% of the injected dose still present in the circulation 5 h after injection (Figure 2c,d). No difference in circulation time was observed between the two fluorophores, indicating that the particles remained intact. Moreover, the circulation kinetics did not significantly differ between the LDV-LNPs and LNPs, nor between individual mice (Appendix A). Thus, surface modification of the LNPs with LDV did not affect their circulation time.

We then investigated in which tissues the LNPs accumulated. To that end, we quantified the fluorescence of LNP Cy5.5 and siRNA Cy7 in organs by fluorescence spectroscopy after 4 (*n* = 4) and 24 h (*n* =2) (Figure 2e–h). LNP- and siRNA-associated fluorescence was found in several organs, including the liver and spleen, and to a smaller extent, in the lungs, heart, kidneys, femora and tibiae. We detected a significantly higher LNPs Cy5.5 signal in the spleen and lungs in animals treated with the LDV-LNPs compared with untargeted LNPs after 4 h (Figure 2e,g). This could be explained by an increased association of LDV-LNPs with VLA-4-positive lymphocytes and macrophages present in these organs [32,44]. In all organs, the overall tissue fluorescence decreased after 24 h. Co-localization of the LNPs Cy5.5 and siRNA Cy7 signal show that the observed biodistribution pattern is unlikely to be caused by Cy7-siRNA leaked from the LNPs, as this would have caused mainly accumulation in the kidneys due to the small size of siRNA (<50 kDa) [45,46,47]. Instead, these data demonstrate LNP-mediated delivery of siRNA to tissues and long bones.

### 3.4. LDV-Decoration Improves LNP Uptake via VLA-4 in Hematopoietic Bone Marrow Cells

To further explore LNPs association with hematopoietic bone marrow cells, we examined the LNP Cy5.5 and Cy7 signal in long bones by whole organ fluorescence spectroscopy and on a single cell level. We observed a high accumulation of both LNPs and siRNA in the femora and tibiae 4 h after injection based on Cy5.5 and Cy7 fluorescence, respectively (Figure 3a, adjusted scaling for the femora and tibiae alone). Although the overall signal decreased over 24 h, the LDV-LNPs were still detectable and were equally distributed along the length of the femora and in the epiphyses of the tibiae (Figure 3b). A significant twofold increase in median LNP Cy5.5 fluorescent signal in single cell suspensions of the bone marrow was observed after 4 and 24 h in the animals treated with LDV-LNPs compared to animals treated with LNPs, indicative of improved accumulation and retention of LNPs in the bone marrow (Figure 3b,c). LNP uptake in the hematopoietic bone marrow cells appeared to be VLA-4-dependent as cells with higher VLA-4 expression showed a significantly higher LNP uptake (Figure 3d,e).

### 3.5. LDV-LNPs Target VLA-4-Positive Immature Myeloid Cells in the Bone Marrow

Leukemia is characterized by the uncontrolled proliferation of immature malignant hematopoietic progenitor-like cells (HSPCs). To investigate whether more immature, progenitor cells can be targeted with the LDV-LNPs and LNPs, we examined LNP uptake and retention in the bone marrow in HSPCs subpopulations. We therefore stained single cells isolated from the bone marrow with antibodies specific for hematopoietic markers such as CD45, CD49d (VLA-4 receptor) and CD34 (expressed by immature myeloid cells). 54% of immature HSPCs (CD45+/VLA-4+/CD34+) were LNP-positive in LDV-LNP-treated mice after 4 and 24 h of injection, as compared to 29% in the LNPs group (Figure 4a–c and Appendix A). Notably, more mature hematopoietic cells (CD34-CD45+CD49d+) also showed enhanced uptake of LDV-LNPs compared to untargeted LNPs (Figure 4b,c: left). Both immature and mature hematopoietic bone marrow cells also showed significant twofold-higher LNP Cy5.5 fluorescence in the LDV-LNPs group compared to the LNPs group, indicating that the improved LDV-LNP uptake is dependent on the expression of the ligand VLA-4 (Figure 4d and Appendix A).

These results clearly show that LDV-LNPs associate with both VLA-4-positive immature and mature HSPCs resulting in higher LNP accumulation and longer retention in the bone marrow. Consequently, these findings strongly support the concept of augmented retain and improved uptake in leukemic cells including leukemic stem cells in vivo.

## 4. Discussion

RNA interference represents a highly attractive alternative to current treatment regimens, however, extrahepatic delivery of siRNA using LNPs has proven difficult due to high liver accumulation. Modification of the LNP surface with targeting ligands could potentially improve LNP uptake beyond the liver. Here we show that targeting the VLA-4 receptor using the LDV motif on the surface of LNPs improved the cellular uptake of LNPs in HSPCs ex vivo and in vivo. These combined data thereby provide evidence for the use of LDV-LNPs for the treatment of hematological disorders.

High liver accumulation is commonly reported for both liposomes and LNP-based therapies. Both LNPs, targeting and non-targeting, used in this study displayed a short circulation time of 45 min. However, longer circulation times are desired as this will improve the passive accumulation of LNPs in extrahepatic tissues [48]. The circulation time is dictated by the lipid components of the nanoparticle formulation and can be finetuned by modifying the length of the PEG-lipid chain, as longer lipid chains will increase the circulation time [49,50]. Since longer circulation times of PEG usually result in reduced endosomal release and will also increase the chances of evoking an anti-PEG immune response [51], PEG does not seem to possess ideal characteristics for being incorporated in LNPs for in vivo usage. As an alternative, PEG could be replaced with polysaccharides, naturally occurring membrane lipids, or stealth-providing compounds such as polysarcosine to shield the nanoparticle from rapid removal by opsonization [52,53,54,55,56]. Additionally, an attractive concept is passive targeting by changing the lipid formulation, however, little is known about the optimal lipid composition for bone marrow targeting [57,58,59]. For liposomes, it is reported that cholesterol aids in bone marrow uptake by phagocytic cells via selective opsonization such as C3 complement protein—an interesting point for future studies on targeting the bone marrow with LNPs [60,61].

Whereas the use of higher dosages to saturate the liver will also improve accumulation in other tissues including the bone marrow, ionizable lipids including D_lin_-MC_3_-DMA show dose-limiting toxicity at higher doses [62,63]. The first FDA- and EMA-approved siRNA LNP treatment, Onpattro, is given at a dose of 0.3 mg/kg body weight once every three weeks. Converting this human dose to the equivalent mouse dose, this is almost double the dose of 2 mg/kg body weight used in this study [64]. Since we already observe a high bone marrow accumulation and retention of the LDV-LNPs at 2 mg/kg body weight, we might already achieve therapeutic effects of the LDV-LNPs in the bone marrow at a clinically safe dose. Thus, efficacy studies in a leukemia disease model would be of high interest. For multiple myeloma, VLA-4-targeting nanoparticles containing chemotherapeutic agents have proven successful in reducing leukemic burden in vivo [31,65]. These studies used immunodeficient mice, which might exhibit longer nanoparticle circulation times due to the lack of immune-related clearance [66,67,68]. In contrast, our model provides evidence that LNPs can reach hematopoietic target cells in the bone marrow even in the presence of a fully functional immune system. Thus, in a murine leukemia model, we expect to observe increased bone marrow accumulation of LDV-LNPs due to improved bone marrow permeability [67,68]. Future studies will uncover whether this is sufficient to reach therapeutically relevant siRNA concentrations in leukemic cells of the bone marrow.

## 5. Conclusions

In summary, we here demonstrate that bone-marrow targeting using LDV-ligand-functionalized LNPs results in substantially enhanced LNP uptake by immature and mature hematopoietic progenitor cells, cell compartments also shown to harbor leukemic stem cells. This study thereby supports the further development of targeted therapeutic interventions for the treatment of leukemia and other hematological disorders.

## Figures and Tables

**Figure 1 pharmaceutics-15-01603-f001:**
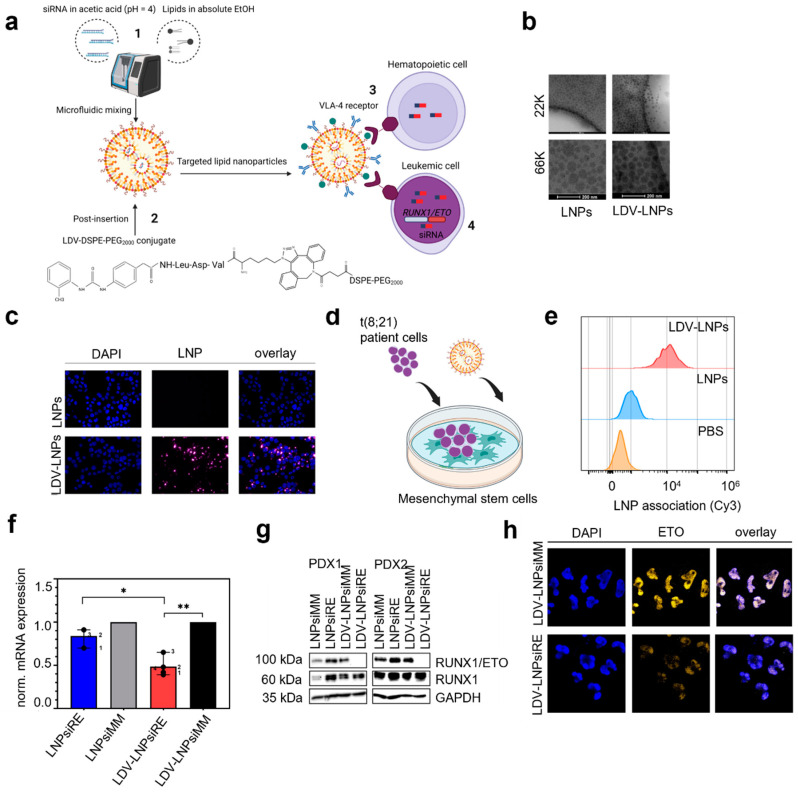
Characterization and siRNA delivery efficacy of lipid nanoparticles in patient-derived leukemia cells. (**a**) Schematic illustration of LDV-LNP production using microfluidic mixing (1), where the LDV-DSPE-PEG_2000_ conjugate is post-inserted into preformed LNPs (2). The LDV-LNPs bind with high affinity to the VLA-4 receptor present on all hematopoietic and leukemic cells (3). In leukemic cells, the active siRNA, siRE-mod, binds to the target mRNA and induces degradation via RNAi (4). (**b**) CryoEM analysis of LNPs (**left**) or LDV-LNPs (**right**) morphology at 22,000× magnification. The bottom panel shows 3-times-enlarged details of the top images. (**c**,**e**–**h**) RUNX1/ETO-expressing AML cells were incubated for 24 h with 2 µg/mL (cell lines) or 4 µg/mL (PDX) siRNA LNPs. (**c**) Uptake of fluorescently labeled LNPs (cyan) without (**top**) and with (**bottom**) LDV-ligand was measured by widefield fluorescence microscopy. Cells were counterstained with DAPI (blue). (**d**) Schematic illustration of the co-culture platform with MSCs feeder layer and RUNX1/ETO-expressing PDX on top. (**e**) LNP Cy3 signal in RUNX1/ETO-expressing PDX after 6 h for LDV-LNPs (red, **top**), LNPs (blue, **middle**) and PBS (orange, **bottom**) as measured by flow cytometry. (**f**,**g**) Reduction of the RUNX1/ETO fusion transcript (**f**) and protein (**g**) of siRNA-LNP-treated PDX detected by qPCR or Western blotting after 3 days of LNP addition. (**h**) Intracellular ETO expression (yellow) was measured by confocal microscopy in RUNX1/ETO-expressing PDX incubated with siMM-mod LDV-LNPs (**top** images) or siRE-mod LDV-LNPs (**bottom** images) Cell were counterstained with DAPI (blue). (**g**) Mean + ranges are displayed. Significance was tested by paired Student’s *t*-test * *p* < 0.05, ** *p* < 0.001, *n* = 4. ^1^ PDX2 bone marrow, ^2^ PDX1 bone marrow, ^3^ PDX2 ascites and ^4^ PDX2 ascites.

**Figure 2 pharmaceutics-15-01603-f002:**
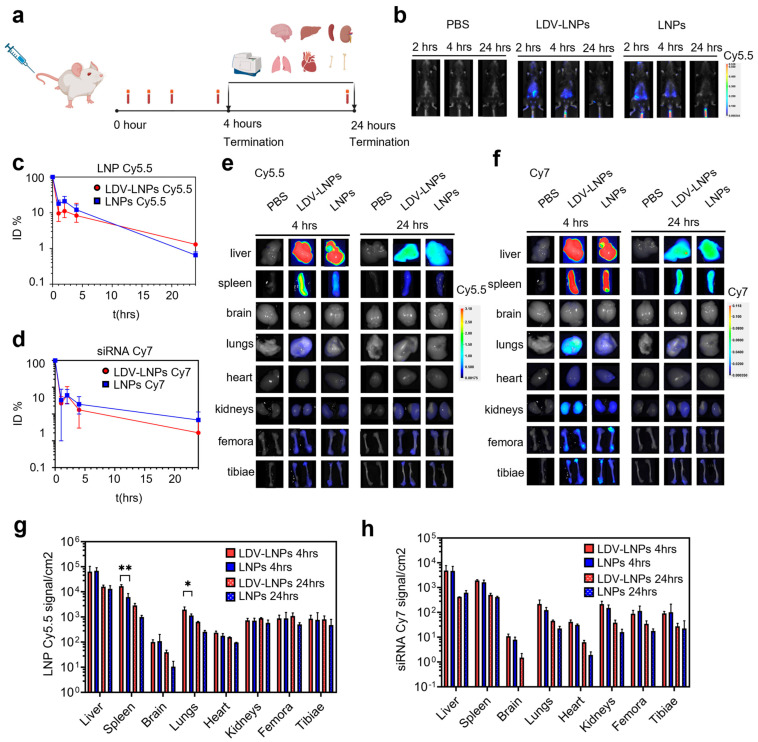
Biodistribution of LDV-LNPs and LNPs after systemic administration. LDV-LNPs or LNPs were administered intravenously at a total dose of 50 ug siRNA per mouse. Mice were live-imaged after 2, 4 and 24 h (**b**) or killed after 4 or 24 h (**c**–**h**). (**a**) Experimental design of the biodistribution study. (**b**) In vivo imaging of LNP-treated mice after 2, 4 and 24 h showing the fluorescent siRNA Cy7 signal in PBS control group (**left**), LDV-LNPs group (**middle**) and LNPs group (**right**). (**c**,**d**) Circulation time of LNPs on a log10 scale. Plasma concentration is expressed as a percentage of the plasma Cy5.5 (**c**) or Cy7 (**d**) fluorescence measured directly after injection (*t* = 1 min). (**e**,**f**) Biodistribution of fluorescently labeled LDV-LNPs and LNPs was measured by whole-organ fluorescence spectroscopy. The image shows the representative LNP Cy5.5 (**e**) fluorescence/white overlay or siRNA Cy7 (**f**) fluorescence/white overlay images of the resected organs 4 and 24 h after injection. (**g**,**h**) Bar graphs showing LNP Cy5.5 (**g**) or siRNA Cy7 (**h**) quantification in whole organs shown in panels e and f using fluorescence spectroscopy. Significance was tested by unpaired Student’s *t*-test * *p* < 0.05, ** *p* < 0.001. (**c**,**d**,**g**,**h**) Mean + ranges are displayed. *n* = 2–6.

**Figure 3 pharmaceutics-15-01603-f003:**
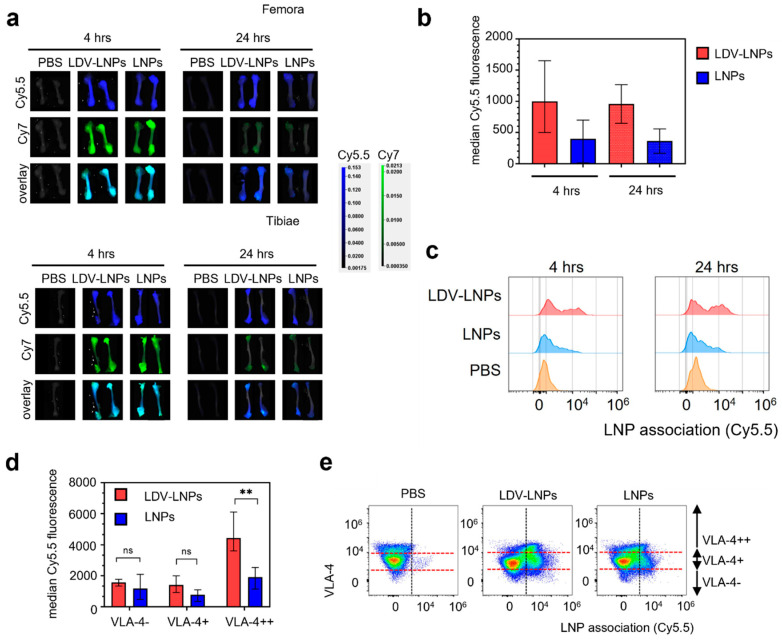
LDV-LNPs and LNPs accumulate in the femora and tibiae. Accumulation of fluorescently labeled LDV-LNPs and LNPs in the femur and tibia was measured at the whole-organ and single-cell levels. (**a**) The LNP Cy5.5 fluorescence/siRNA Cy7/white overlay images for the femora (**top** figure) and tibiae (**bottom** figure) 4 and 24 h after injection. The scale of both channels has been adjusted for the long bones alone. (**b**) Median Cy5.5 fluorescence (minus the Cy5.5 fluorescence signal in control PBS) in the bulk single femur cells as determined by flow cytometry. (**c**) LNP signal in bulk single femur cells after 4 (**left**) or 24 h (**right**) for LDV-LNPs (red, **top**), LNPs (blue, **middle**) and PBS (orange, **bottom**) as measured by flow cytometry. (**d**) Median Cy5.5 fluorescence in VLA-4-positive femur cells after 4 h as determined by flow cytometry. (**e**) Scatter plots showing the VLA-4 expression and LNP Cy5.5 accumulation in single femur cells after 4 h for PBS, LDV-LNPs and LNPs as determined by multiparameter flow analysis. The red dotted lines indicate gating for VLA-4-, VLA-4+ and VLA-4++ cells. (**b**,**d**) Mean + ranges are displayed. *n* = 4 (4 h) *n* = 2 (24 h). Significance was tested by two-tailed unpaired Student’s *t*-test, ** *p* < 0.001, ns = not significant, unpaired two-tailed Student’s *t*-test.

**Figure 4 pharmaceutics-15-01603-f004:**
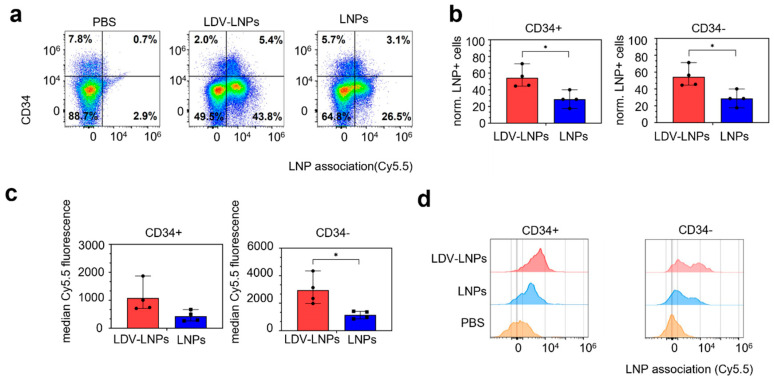
Increased uptake of LDV-LNPs in HPSC in the bone marrow. (**a**) Scatter plots showing CD34 expression and LNP association with CD45+/VLA-4+ cells in the bone marrow after 4 h after administration of PBS, LDV-LNPs, or LNPs as determined by multiparameter flow analysis. (**b**) Bar graphs showing the percentage of LNP+ cells in CD45+/VLA-4+/CD34+ (**left**) or CD45+/VLA-4+/CD34- (**right**) bone marrow cells as determined by multiparameter flow analysis after 4 h. (**c**,**d**) LNP Cy5.5 association with CD34+ (**left**) and CD34- (**right**) bone marrow cells after 4 h displayed in bar graphs (**c**) and histograms (**d**) where LDV-LNPs is red, LNPs blue and control PBS orange. (**b**,**c**) Means + ranges are displayed. Significance was tested by two-tailed unpaired Student’s *t*-test * *p* < 0.05, unpaired two-tailed Student’s *t*-test. *n* = 4.

**Table 1 pharmaceutics-15-01603-t001:** Characteristics of LNPs used in this study. DLS measurement of LNPs. mean +/− range is displayed (*n* = 3). For LNP and LDV-LNP, three independent batches of LNPs have been prepared. For LNP-Cy7 and LDV-LNP-Cy7, the displayed ranges indicate technical triplicates.

Formulation	Components	Particle Size (nm)	PDI	Zeta Potential (mV)	Encapsulation Efficiency (%)
LNP	Dlin-MC3-DMA/DSPC/Cholesterol/DMG-PEG_2000_ = 50/10/38.5/1.5	66 (57, 83)	0.14 (0.12, 0.16)	−2.5 (−4.2, −1.0)	94 (93, 95)
LDV-LNP	Dlin-MC3-DMA/DSPC/Cholesterol/DMG-PEG_2000_/LDV-azide-DBCO-DSPE-PEG_2000_ = 50/10/38.5/1.5/0.1	94 (87, 105)	0.26 (0.20, 0.35)	−4.1 (−6.3, −2.3)	93 (91, 94)
LNP-Cy7	Dlin-MC3-DMA/DSPC/Cholesterol/DMG-PEG_2000_/DBCO-DSPE-PEG_2000_ = 50/10/38.5/1.5/0.1	69 (68, 70)	0.08 (0.07, 0.09)	−0.3 (−0.2, −0.3)	97
LDV-LNP-Cy7	Dlin-MC3-DMA/DSPC/Cholesterol/DMG-PEG_2000_/LDV-azide-DBCO-DSPE-PEG_2000_ = 50/10/38.5/1.5/0.1	74 (73, 75)	0.08 (0.07, 0.09)	−3.1 (−2.7, −3.7)	97

## Data Availability

All created data is available in this manuscript and the Appendix A.

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
