# Peer review of "Increased Bone Marrow Uptake and Accumulation of Very-Late Antigen-4 Targeted Lipid Nanoparticles"

_pharmaceutics, 2023, doi:10.3390/pharmaceutics15061603_

Round 1
Reviewer 1 Report
This manuscript explored the tripeptide targeted LNP formulations for siRNA delivery to hematopoietic progenitor cells in bone marrow. The cellular uptake and retention of LNP/siRNA was evaluated using patient-derived leukemia cells. Additionally the pharmacokinetics and biodistribution were assessed in mice model. It is a very interesting research to explore new therapeutic strategies to target hematological diseases. The research theme fit into the scope of journal. Experimental design and methodology used are sound and the reasoning is logic.
Minor:
1. Dio-labelled LNP was used for biodistribution, while cy5.5 was used for circulation studies, any reason for choosing two different dye? where is the results of Dio-LNP?
2. In section3.3, quote” In vivo imaging of C57Bl/6J mice showed highest siRNA Cy7 fluorescent signal in the liver region after 4 hours, with a strong reduction of the signal after 24 hours (Fig. 2b)”. wasn’t figure2b showing the Cy5.5-labbelled LNP?
3. In section3.3 quote “ These combined results show that the observed biodistribution pattern is not caused by unconjugated Cy5.5 leaked from LNPs nor naked Cy7-siRNA, as this would have caused mainly accumulation in the kidneys due to the small size of siRNA and/or fluorophores (< 50 kDa)”. However, there showed a strong Cy7 signal in kidney at 4 hrs for both LDV-LNP and LNP in figure2f. Is it because of siRNA dissociation from LNP? because the encapsulation efficiency is great than 93%, the free siRNA shall come from the leakage from LNP.
Author Response
Reviewer 1
This manuscript explored the tripeptide targeted LNP formulations for siRNA delivery to hematopoietic progenitor cells in bone marrow. The cellular uptake and retention of LNP/siRNA was evaluated using patient-derived leukemia cells. Additionally the pharmacokinetics and biodistribution were assessed in mice model. It is a very interesting research to explore new therapeutic strategies to target hematological diseases. The research theme fit into the scope of journal. Experimental design and methodology used are sound and the reasoning is logic.
Minor:
Dio-labelled LNP was used for biodistribution, while cy5.5 was used for circulation studies, any reason for choosing two different dye? where is the results of Dio-LNP?
We changed the LNP label to Cy5.5 for the circulation studies because we wanted to colocalize LNP label with the siRNA label in the whole organs. This was not possible with the DiO as LNP label due to the settings of the PEARL imager. We therefore did not include results with the DiO dye and do not mention this dye anymore in the Methods.
Section 2.13 now reads: ‘The biodistribution of LDV-LNPs and LNPs was assessed in female C57/Bl6J (n = 8, weight between 18 and 22 g, 11 weeks old, Charles River, Leiden, the Netherlands) upon tail vein injection of LNPs loaded with Cy7-labelled siRNA (50% of total siRNA content).’
- In section3.3, quote” In vivo imaging of C57Bl/6J mice showed highest siRNA Cy7 fluorescent signal in the liver region after 4 hours, with a strong reduction of the signal after 24 hours (Fig. 2b)”. wasn’t figure2b showing the Cy5.5-labbelled LNP?
We thank the reviewer for carefully reading the manuscript and pointing this out. It should be Cy7 as the siRNA was conjugated to that fluorophore. We have changed Cy5.5 in the legend of Fig. 2b to Cy7.
In section3.3 quote “ These combined results show that the observed biodistribution pattern is not caused by unconjugated Cy5.5 leaked from LNPs nor naked Cy7-siRNA, as this would have caused mainly accumulation in the kidneys due to the small size of siRNA and/or fluorophores (< 50 kDa)”. However, there showed a strong Cy7 signal in kidney at 4 hrs for both LDV-LNP and LNP in figure2f. Is it because of siRNA dissociation from LNP? because the encapsulation efficiency is great than 93%, the free siRNA shall come from the leakage from LNP.
There is indeed a strong signal for the Cy7 siRNA label in the kidney at 4 hours, but there is also a strong signal for the Cy5.5 LNP label. This is indicative of colocalization of the siRNA and the LNP in the whole organs. The measured circulation time for both the Cy5.5 and Cy7 label was the same (Fig. 2c, 2d), indicating that the particles remained intact in the blood stream. Therefore, we do not think that the Cy7 fluorescent signal in the kidneys is caused by leakage of siRNA from the LNPs. We have now emphasized this in the manuscript.
Section 3.3 now reads: ‘LNP and siRNA-associated fluorescence was found in several organs, including the liver, spleen and to a smaller extent in the lungs, heart, kidneys, femora and tibiae. We detected a significantly higher LNPsCy5.5 signal in the spleen and lungs in animals treated with the LDV-LNPs compared with untargeted LNPs after 4 hours (Fig. 2e, 2g). This could be explained by an increased association of LDV-LNPs with VLA-4 positive lymphocytes and macrophages present in these organs (1, 2). In all organs the overall tissue fluorescence decreased after 24 hours. Co-localization of the LNPs Cy5.5 and siRNA Cy7 signal show that the observed biodistribution pattern is unlikely to be caused by Cy7-siRNA leaked from the LNPs, as this would have caused mainly accumulation in the kidneys due to the small size of siRNA (< 50 kDa) (3-5). Instead, these data demonstrate LNP-mediated delivery of siRNA to tissues and long bones.’

Reviewer 2 Report
Manuscript by Swart et al. deals with the highly actual and important issue, namely, the LNP-mediated targeted siRNA delivery to the bone marrow hematopoietic progenitor-like cells. Authors provided compelling proof-of-concept for delivery of siRNA to the bone marrow via LNP surface modification by LDV tripeptide, which binds to very-late antigen-4 receptor. VLA-4 is abundantly expressed on the hematopoietic stem and progenitor cells in the bone marrow. Also, they demonstrated LNP-mediated specific silencing of the pathological RUNX1/ETO gene (both on RNA and protein level) in leukemic blasts and in patient-derived xenotransplants, as an in vitro leukemia model.
The paper has multiple merits, such as clarity and persuasiveness of the data, high methodical level, excellent presentation of results, and, in sum, strong scientific impact. Surely, it can be published in the Pharmaceutics after some minor (mostly technical) correction.
I have a few remarks to be addressed.
1. In the introductory section, authors write about the primary organs of the reticuloendothelial system (RES): liver, spleen and lungs, as main places of LNPs accumulation. As far as I know, the term “reticuloendothelial system” is rather obsolete (though still in use sometimes), and means the complex of endothelium and tissue monocytes capable to the endocytosis. Further, authors mention hepatocytes as most active LNPs uptakers. Definitely, hepatocytes do not belong to the RES, and this statement is rather misleading. In order to avoid confusion, I would suggest to exclude any RES mention from the paper, especially since RES has no relation at all to the further experimental data.
2. Section 2.1. «As a mismatch control, we used siMM-mod, where two nucleotides in the siRE sequence have been swapped». At first, it would be good to specify which nucleotides were swapped. Then, I am afraid, this is not a real mismatch. The two nucleotide substitutions are rather insufficient to stop the target mRNA sequence binding, and this siRNA is apparently still capable to the partial gene silencing (similarly to how endogenous miRNAs do). So, I suggest, authors' negative control data on Fig. 1f and 1g are in fact a partial gene silencing data. As a negative control, it is better to use a scramble siRNA, which indeed has no mRNA sequence for the recognition. I suppose, it would be good to discuss somehow this point in the paper.
3. Section 2.10. «qPCR primers are provided in suppl. Table 2». There is no table 2 in the supplementary materials (at least in the file which I obtained). Please check this.
4. Figure 1b. Cryo-EM. There is no mention of this data in the 3.2 Results text. I think, some words related to this should be added.
5. While Figure 1 is described correctly in the 3.2 text section (apart from the above number 4 comment), there is the total mess in the legend to Fig.1. “(c) Size variability LNPs, LDV-LNPs (n = 3, produced over a period of 1 year), LNPs-Cy7 and LDV-LNPs Cy7”. This image is lacking. Respectively, legends to all other parts of the Fig.1 (c-h) are shifted. This has to be corrected.
6. Fig. S1 from the supplementary is not mentioned in the text. Please write something about it, or delete it.
Of course, the above remarks do not detract from the great merits of the article. It was a big pleasure for me to read and review it.
Author Response
Reviewer 2
Manuscript by Swart et al. deals with the highly actual and important issue, namely, the LNP-mediated targeted siRNA delivery to the bone marrow hematopoietic progenitor-like cells. Authors provided compelling proof-of-concept for delivery of siRNA to the bone marrow via LNP surface modification by LDV tripeptide, which binds to very-late antigen-4 receptor. VLA-4 is abundantly expressed on the hematopoietic stem and progenitor cells in the bone marrow. Also, they demonstrated LNP-mediated specific silencing of the pathological RUNX1/ETO gene (both on RNA and protein level) in leukemic blasts and in patient-derived xenotransplants, as an in vitro leukemia model.
The paper has multiple merits, such as clarity and persuasiveness of the data, high methodical level, excellent presentation of results, and, in sum, strong scientific impact. Surely, it can be published in the Pharmaceutics after some minor (mostly technical) correction.
I have a few remarks to be addressed.
- In the introductory section, authors write about the primary organs of the reticuloendothelial system (RES): liver, spleen and lungs, as main places of LNPs accumulation. As far as I know, the term “reticuloendothelial system” is rather obsolete (though still in use sometimes), and means the complex of endothelium and tissue monocytes capable to the endocytosis. Further, authors mention hepatocytes as most active LNPs uptakers. Definitely, hepatocytes do not belong to the RES, and this statement is rather misleading. In order to avoid confusion, I would suggest to exclude any RES mention from the paper, especially since RES has no relation at all to the further experimental data.
We have changed the introduction to: ‘The majority of systemically administered LNPs accumulate in the liver due to its large size, functionalized vascular structure and for some LNP types surface-absorbed apolipoprotein E mediated uptake by the hepatocytes (6-8). The resultant liver accumulation is widely exploited to effectively modulate therapeutic targets in this organ. Targeting other organs and tissues has proven to be more challenging (7, 9, 10).’
- Section 2.1. «As a mismatch control, we used siMM-mod, where two nucleotides in the siRE sequence have been swapped». At first, it would be good to specify which nucleotides were swapped. Then, I am afraid, this is not a real mismatch. The two nucleotide substitutions are rather insufficient to stop the target mRNA sequence binding, and this siRNA is apparently still capable to the partial gene silencing (similarly to how endogenous miRNAs do). So, I suggest, authors' negative control data on Fig. 1f and 1g are in fact a partial gene silencing data. As a negative control, it is better to use a scramble siRNA, which indeed has no mRNA sequence for the recognition. I suppose, it would be good to discuss somehow this point in the paper.
We have added the sequence of the mismatch siRNA control in Table S1. We do not observe partial gene silencing with the mismatch siRNA; therefore we think that it is a good negative control. We have now added Fig. S2 where we normalized a subset of the samples from Fig. 1f to the ddCt value of the mock. The mRNA expression does not significantly differ between the mock and LDV-LNPsiMM cells; thus the mismatch siRNA does not reduce the RUNX1/ETO transcript and does provide a proper negative control in these experiments.
Figure S2: Reduction of RUNX1/ETO transcript in PDX. RUNX1/ETO-expressing AML cells were incubated for 24 hours with 4 µg/ml siRNA LNPs followed by qPCR analysis at day 3. This figure displays a subset of the samples from Fig. 1f, samples are here normalized against the ddCt of the mock cells. Mean + ranges are displayed. Significance was tested by paired Student’s t-test *P <0.05, **P < 0.001, ***P <0.001, n = 3.
The results section 3.2 now reads: ‘The uptake of LDV-LNPs by RUNX1/ETO-expressing PDX was increased 10-fold compared to untargeted LNPs within 24 hours and led to a twofold reduction of the RUNX1/ETO transcript and a complete loss of the fusion protein in different PDX samples after 3 days as compared to the controls (Fig. 1e – g, Suppl. Fig. 2).’
Section 2.10. «qPCR primers are provided in suppl. Table 2». There is no table 2 in the supplementary materials (at least in the file which I obtained). Please check this.
We thank the reviewer for noticing this. We have now added the primer sequences to supplemental table 2.
- Figure 1b. Cryo-EM. There is no mention of this data in the 3.2 Results text. I think, some words related to this should be added.
We have now mentioned the cryo-EM data in Results section 3.1 as suggested:
‘Both LDV-conjugate post-inserted LNPs or LNPs containing fluorescent-labelled siRNA and DSPE-Cy5.5 displayed similar physicochemical characteristics with a hydrodynamic diameter <100 nm (Table 1), PDI of 0.2 and slightly negative zeta potential (Table 1). CryoEM imaging confirmed that LNPs and LDV-LNPs are uniformly sized (Fig. 2b).’
- While Figure 1 is described correctly in the 3.2 text section (apart from the above number 4 comment), there is the total mess in the legend to Fig.1. “(c) Size variability LNPs, LDV-LNPs (n = 3, produced over a period of 1 year), LNPs-Cy7 and LDV-LNPs Cy7”. This image is lacking. Respectively, legends to all other parts of the Fig.1 (c-h) are shifted. This has to be corrected.
We apologize for this omission and have adjusted the figure numbering of Figure 1. The legend of Fig. 1 now reads:
‘(a) Schematic illustration of LDV-LNP production using microfluidic mixing (1), where LDV-DSPE-PEG2000 conjugate is post-inserted into preformed LNPs (2). The LDV-LNPs bind with high affinity to the VLA-4 receptor present on all hematopoietic and leukemic cells (3). In leukemic cells the active siRNA, siRE, binds to the target mRNA and induces degradation via RNAi (4). (b) CryoEM analysis of LNPs (left) or LDV-LNPs (right) morphology at 22,000x magnification. The bottom panel shows 3 times enlarged details of the top images. (c, e-h) RUNX1/ETO-expressing AML cells were incubated for 24 hours with 2 µg/ml (cell lines) or 4 µg/ml (PDX) siRNA LNPs. (c) Uptake of fluorescently labeled LNPs (cyan) without (top) and with (bottom) LDV-ligand was measured by widefield fluorescence microscopy. Cells were counterstained with DAPI (blue). (d) Schematic illustration of the co-culture platform with MSCs feeder layer and RUNX1/ETO-expressing PDX on top. (e) LNP Cy3 signal in RUNX1/ETO-expressing PDX after 6 hours for LDV-LNPs (red, top), LNPs (blue, middle) and PBS (orange, bottom) as measured by flow cytometry. (f, g) Reduction of the RUNX1/ETO fusion transcript (f) and protein (g) of siRNA LNPs treated PDX detected by qPCR or western blotting after 3 days of LNP addition. (h) Intracellular ETO expression (yellow) was measured by confocal microscopy in RUNX1/ETO-expressing PDX incubated with siMM-mod LDV-LNPs (top images) or siRE-mod LDV-LNPs (bottom images) Cell were counterstained with DAPI (blue). (g) Mean + ranges are displayed. Significance was tested by paired Student’s t-test *P <0.05, **P < 0.001, ***P <0.001, n = 4. 1 PDX2 bone marrow, 2 PDX1 bone marrow, 3 PDX2 ascites and 4 PDX2 ascites.’
- Fig. S1 from the supplementary is not mentioned in the text. Please write something about it, or delete it.
The reference to Fig. S1 is made in the methods section 2.15 Flow cytometry:
‘Data were analyzed using FlowJo.v10.7.1. Cell populations were identified according to gating strategies shown in Suppl. Fig. 1.’
Of course, the above remarks do not detract from the great merits of the article. It was a big pleasure for me to read and review it.
Reviewer 3 Report
Dear Authors
The article “Increased Bone Marrow Uptake and Accumulation of Very-Late Antigen-4 Targeted Lipid Nanoparticles” is very interesting and well-designed. Following are a few suggestions which might be helpful for the improvement of the article a bit
· Why are LNPs different from other lipid nanoparticles? In the introductory section, authors might discuss the advantages of LNPs over other lipid particles.
· • The DLin-MC3 has a pKa value of 6.2-6.4. The authors should measure the zeta potential at that pH to validate the complexation.
· One of the most crucial parameters in this paper is the modification of the LNPs lipids. The author should describe a thorough methodology as well as process parameters for the Click chemistry conjugation and functionalization of LNPs.
· Correct “H20” to H2O
· Animal per group is very small for biodistribution study. Even though three is a statistically recognized quantity, utilizing three animals per group does not produce reliable findings. Authors should justify the use of “randomized into groups of three mice”
· Authors should provide the gel electrophoresis data of the n/p ratio of the complexation for better understanding. Furthermore, the authors demonstrated in Table 1 that a negative zeta potential indicated the presence of free siRNA. The authors should clarify why DV-LNP-Cy7 has a zeta potential of -3.1 in order to fully understand the complexation of the n/p ratio.
· Figure 1 b. The authors mention the magnification of 29K whereas in the image it's 22 K. Authors should take a look at the magnification.
· Authors should correct the figure legend of Figure 1, as they do not match with the figure numbering
· Which Cyanine (Cy3/Cy5.5/ Cy7) used for the study? And why authors used different Cyanine for different studies?
· What is the IC50 value of the LNPs formulation?
· Provide a clear and large image of In vivo imaging of LNP-treated mice in Figure 2b.
· Rewrite the conclusion to better support the study claim and outcome.
The article needed through spelling checks
Author Response
Reviewer 3
Dear Authors
The article “Increased Bone Marrow Uptake and Accumulation of Very-Late Antigen-4 Targeted Lipid Nanoparticles” is very interesting and well-designed. Following are a few suggestions which might be helpful for the improvement of the article a bit
- Why are LNPs different from other lipid nanoparticles? In the introductory section, authors might discuss the advantages of LNPs over other lipid particles.
We have added the following sentences to the introduction:
‘Because of poor pharmacokinetic properties of siRNAs, caused by low stability, poor uptake by cells, fast clearance and induction of immunogenic responses, delivery vehicles such as micelles, liposomes, nanoplexes or lipid nanoparticles (LNPs) are required (11-20). LNPs are currently amongst the most promising non-viral delivery systems used for the delivery of siRNAs, because of their high encapsulation efficiency combined with reduced immunogenicity and improved circulation times (19).’
- •The DLin-MC3 has a pKa value of 6.2-6.4. The authors should measure the zeta potential at that pH to validate the complexation.
For measuring the zetapotential we have followed the EU NCL guidelines ‘Characterisation of nanomaterials’ where is stated that the zetapotential should be measured in 10 mM NaCl or similar buffer. We choose to measure at pH 7.5, as this is the commonly accepted standard in literature.
- One of the most crucial parameters in this paper is the modification of the LNPs lipids. The author should describe a thorough methodology as well as process parameters for the Click chemistry conjugation and functionalization of LNPs.
The detailed methodology and the process parameters for the click chemistry conjugation can be found in a previously published manuscript: Swart LE, Koekman CA, Seinen CW, Issa H, Rasouli M, Schiffelers RM, Heidenreich O. A robust post-insertion method for the preparation of targeted siRNA LNPs. Int J Pharm. 2022 May 25;620:121741. doi: 10.1016/j.ijpharm.2022.121741. Epub 2022 Apr 11. PMID: 35421533.
We have now added a summary of this click chemistry method in methods section 2.6, for more details we refer to the manuscript:
‘Functionalized LNPs were prepared following the protocol as we previously established (21). In brief, Cyanine 3-azide (Cy3) or LDV-azide were first conjugated to the ring-constrained alkyne dibenzocyclooctyne (DBCO) which is covalently linked to DSPE-PEG2000 overnight at room temperate at a 1:3 molar ratio. The conjugated product was immediately post-inserted at a 0.1% molar ratio into preformed LNPs for 30 min at 45 °C (21). Functionalized LNPs were kept at 4°C protected from light for up to 7 days.’
- Correct “H20” to H2O
We have corrected this.
- Animal per group is very small for biodistribution study. Even though three is a statistically recognized quantity, utilizing three animals per group does not produce reliable findings. Authors should justify the use of “randomized into groups of three mice”
Although we understand the concern of this reviewer, we do not think that three animals per group is too small for a biodistribution study. The variability in biodistribution between wild-type animals is very low, commonly reported in literature and demonstrated in our own data by the low range between the animals. Increasing the sample size here would have not provided additional information. Following the “3R” principles, we therefore decided to choose the minimum amount of animals per group that would provide us a statistically significant difference as determined by ANOVA: Fixed effects, omnibus, one-way a priori testing in the G.Power 3.1 software.
- Authors should provide the gel electrophoresis data of the n/p ratio of the complexation for better understanding. Furthermore, the authors demonstrated in Table 1 that a negative zeta potential indicated the presence of free siRNA. The authors should clarify why DV-LNP-Cy7 has a zeta potential of -3.1 in order to fully understand the complexation of the n/p ratio.
Upon neutralization, all our LNP formulations adopt a slightly negative zeta potential, a change commonly noticed and described in literature (e.g. Tarab-Ravski et al, 2023, Adv. Sci). This might be due to the phosphocholine residues which adopt a slightly negative overall charge at neutral pH (Monicelli et al, 1994, Biophys. J.). However, determining free siRNA using the Ribogreen assay showed encapsulation values above 90%. Thus, one cannot exclude that siRNA bound to the LNP surface might contribute to this negative charge. Nevertheless, this high degree of encapsulation also demonstrates that the N:P ratio of 4 widely used for the DLin-MC3-DMA – siRNA combinations is fully sufficient for complexation of the cargo.
- Figure 1 b. The authors mention the magnification of 29K whereas in the image it's 22 K. Authors should take a look at the magnification.
We used 22,000x magnification, this is changed in the manuscript now.
- Authors should correct the figure legend of Figure 1, as they do not match with the figure numbering
We apologize for this unintended omission and have adjusted the figure numbering of Figure 1. The legend of Fig. 1 now reads:
‘(a) Schematic illustration of LDV-LNP production using microfluidic mixing (1), where LDV-DSPE-PEG2000 conjugate is post-inserted into preformed LNPs (2). The LDV-LNPs bind with high affinity to the VLA-4 receptor present on all hematopoietic and leukemic cells (3). In leukemic cells the active siRNA, siRE, binds to the target mRNA and induces degradation via RNAi (4). (b) CryoEM analysis of LNPs (left) or LDV-LNPs (right) morphology at 22,000x magnification. The bottom panel shows 3 times enlarged details of the top images. (c, e-h) RUNX1/ETO-expressing AML cells were incubated for 24 hours with 2 µg/ml (cell lines) or 4 µg/ml (PDX) siRNA LNPs. (c) Uptake of fluorescently labeled LNPs (cyan) without (top) and with (bottom) LDV-ligand was measured by widefield fluorescence microscopy. Cells were counterstained with DAPI (blue). (d) Schematic illustration of the co-culture platform with MSCs feeder layer and RUNX1/ETO-expressing PDX on top. (e) LNP Cy3 signal in RUNX1/ETO-expressing PDX after 6 hours for LDV-LNPs (red, top), LNPs (blue, middle) and PBS (orange, bottom) as measured by flow cytometry. (f, g) Reduction of the RUNX1/ETO fusion transcript (f) and protein (g) of siRNA LNPs treated PDX detected by qPCR or western blotting after 3 days of LNP addition. (h) Intracellular ETO expression (yellow) was measured by confocal microscopy in RUNX1/ETO-expressing PDX incubated with siMM-mod LDV-LNPs (top images) or siRE-mod LDV-LNPs (bottom images) Cell were counterstained with DAPI (blue). (g) Mean + ranges are displayed. Significance was tested by paired Student’s t-test *P <0.05, **P < 0.001, ***P <0.001, n = 4. 1 PDX2 bone marrow, 2 PDX1 bone marrow, 3 PDX2 ascites and 4 PDX2 ascites.’
- Which Cyanine(Cy3/Cy5.5/ Cy7) used for the study? And why authors used different Cyanine for different studies?
We used different fluorophores for the ex vivo and in vivo imaging, because of (1) the available equipment and (2) the autofluorescence of the organs. For ex vivo uptake studies we have used Cy3 for optimal detection by our microscopy and flow cytometric setup. For in vivo studies we chose Cy5.5 as LNP label and conjugate Cy7 to the siRNA to minimize interference by autofluorescence.
We have emphasized this in Methods 2.1 siRNA LNPs preparation: ‘For ex vivo visualization 0.1% molar ratio Cy3-DBCO-DSPE-PEG2000 was post inserted into preformed LNPs. For in vivo visualization 50% or 30% of the siRE-mod was replaced by siRE-mod-Cy7 (Axolabs, Kulmbach, Germany), conjugated to the sense strand and 0.2% molar ratio DSPE-Cy5.5 was added as lipid label.’
- What is the IC50 value of the LNPs formulation?.
This depends on the readout and the sample. For patient material the cells numbers are limited, because of availability and their low expansion capacity ex vivo, therefore we choose a concentration for the LNPs for the biodistribution study that was based on the well tolerated concentration we used in vivo for non-targeted LNPs (please see reference 5 in the manuscript).
Provide a clear and large image of In vivo imaging of LNP-treated mice in Figure 2b.
We have enlarged Fig. 2b as the reviewer suggested.
- Rewrite the conclusion to better support the study claim and outcome.
We have modified the conclusion as suggested by the reviewer, the conclusion now reads:
‘In summary, we here demonstrate that bone marrow targeting using LDV-ligand functionalized LNPs results in substantially enhanced LNP uptake by immature and mature hematopoietic progenitor cells, cell compartments also shown to harbor leukaemic stem cells. This study thereby supports the further development of targeted therapeutic interventions for the treatment of leukemia and other hematological disorders.’
Round 2
Reviewer 3 Report
Check the grammar and spelling before final submission
Minor spelling and grammar check needed